# *Trans* Fatty Acids in Human Milk in Latvia: Association with Dietary Habits during the Lactation Period

**DOI:** 10.3390/nu13092967

**Published:** 2021-08-26

**Authors:** Līva Aumeistere, Alīna Beluško, Inga Ciproviča, Dace Zavadska

**Affiliations:** 1Faculty of Food Technology, Latvia University of Life Sciences and Technologies, Rīgas iela 22, LV-3004 Jelgava, Latvia; alinapolipartova@gmail.com (A.B.); inga.ciprovica@llu.lv (I.C.); 2Department of Pediatrics, Riga Stradiņš University, Vienības Gatve 45, LV-1004 Riga, Latvia; dace.zavadska@rsu.lv

**Keywords:** *trans* fatty acids, human milk, dietary habits, nutrition

## Abstract

The human milk fatty acid, including *trans* fatty acid, composition is predominantly affected by the maternal diet. The aim of this research was to determine the *trans* fatty acid level in human milk among lactating women in Latvia, and to evaluate how maternal dietary habits affect the *trans* fatty acid composition of human milk. In total, 70 lactating women participated in this cross-sectional study. A 72-hour food diary and food frequency questionnaire were used to evaluate maternal dietary habits. Different *trans* fatty acids in human milk samples were determined using gas chromatography (Agilent 6890N, Agilent Technologies Incorporated, the United States). Overall, the dietary intake of *trans* fatty acids among the participants was 0.54 ± 0.79 g per day. The total *trans* fatty acid level in the human milk samples was 2.30% ± 0.60%. The composition of *trans* fatty acids found in human milk was associated with maternal dietary habits. Higher elaidic acid, vaccenic acid and total *trans* fatty acid levels in human milk were found among participants with a higher milk and dairy product intake. Meat and meat product intake were associated with a higher vaccenic acid and total *trans* fatty acid levels in human milk. A moderate association was also established between maternal *trans* fatty acid intake and the total *trans* fatty acid level in human milk. The obtained correlations indicate that maternal dietary habits during lactation can impact the composition of *trans* fatty acids found in human milk.

## 1. Introduction

Approximately half of infants in Latvia receive human milk for the first six months of life, and for 15–20% of them, human milk is the only nutrient source during this time [1]. Nutrients for human milk are derived from maternal body stores and those absorbed directly from the maternal diet [2]. Therefore, a well-balanced diet during lactation is important, both for the mother and breastfed infant [2].

There are the following two sources of fatty acids in human milk depending on the length of fatty acids:fatty acids up to 14 carbon atoms are synthesised from the glucose via pentose phosphate cycle within the mammary glands;long-chain fatty acids (≥C16) are derived from the maternal bloodstream, and they are transported into lactocytes via the protein-mediated membrane transport system [2].

Therefore, the fatty acid (≥C16) composition of human milk is variable and impacted by maternal dietary habits during lactation [2]. For example, we have previously reported that fish intake reflects on the docosahexaenoic acid level in human milk among lactating women from Latvia [3].

Human milk can also contain a variable amount of *trans* fatty acids, which may come from the following two dietary sources:natural—via biohydrogenation in the rumen (meat, milk and dairy products). Dominant *trans* fatty acids from natural food sources are vaccenic acid (C18:1 n11*t*) and conjugated linoleic acid (main isomer—rumenic acid, C18:2 n9*c*, n11*t*);industrially—via frying, grilling or the hydrogenation of oils (confectionary, bakery goods, French fries, etc.). The dominant *trans* fatty acids from industrially produced foodstuff are elaidic acid (C18:1 n9*t*) and linolelaidic acid (C18:2 n9*t*, n12*t*) [4,5].

*Trans* fatty acids can have a harmful effect on the development of an infant. A higher *trans* fatty acid level in the body interferes with the desaturation of linoleic acid and α-linolenic acid [6]. This could lead to lower levels of arachidonic acid and docosahexaenoic acid in the body, respectively. Both are important fatty acids for the normal development of the brain and eyesight [6]. These negative effects are associated with *trans* fatty acids that are derived from partially hydrogenated vegetable oils or grilled, fried foods [7]. Opposingly, natural *trans* fatty acids are associated with health-related benefits [8]. Some researchers have observed that a higher level of natural *trans* fatty acids in human milk was related to a lower risk of atopic dermatitis, eczema and food allergies for the infant [9]. A possible explanation of this observation is that natural *trans* fatty acids, such as conjugated linoleic acid, modulate immune function in the human body by reducing the production of pro-inflammatory mediators (cytokines, prostaglandins, etc.) [10].

The current general position is that the human body cannot synthesise *trans* fatty acids; therefore, the only source for *trans* fatty acids found in human milk can be the maternal diet [11], but some studies indicate that vaccenic acid can be converted to rumenic acid (C18:2 n9*c*, n11*t*) by Δ9-desaturase in the mammary glands during lactation [5,12].

Current legislation for the Member States of the European Union, including Latvia [13,14], declares the maximum permitted amount of *trans* fatty acids in food products other than the *trans* fatty acids naturally occurring in fat of animal origin. The maximum permitted amount shall not exceed 2 g per 100 g of fat.

As human milk is the sole source of nutrition for 15–20% of infants in Latvia in the first six months of life [1], the quality of human milk is of great importance. Therefore, the aim of this research was to determine the *trans* fatty acid level in human milk among lactating women in Latvia, and to evaluate how maternal dietary habits affect the *trans* fatty acid composition of human milk.

## 2. Materials and Methods

This cross-sectional study was conducted during the period January till December year 2020. The inclusion criteria for participants were as follows:a signed consent form;residing in Latvia during the study period;singleton pregnancy;at least one month postpartum;exclusively or partially breastfeeding.

The study was conducted in accordance with the Declaration of Helsinki, and prior, approval from the Riga Stradinš University Ethics Committee was obtained (protocol code 6-1/01/6). Written consent was obtained from all the participants before the study. Exclusion criteria for the study were unsigned consent form and/or non-compliance with the inclusion requirements.

In total, 70 participants completed the study. Initially, 108 participants met the study inclusion criteria; however, 38 dropped out.

### 2.1. Collection of Human Milk Samples, Food Data and Other Information

Women were invited to participate in the study using a poster published on social media member groups for lactating mothers. If a woman met the study inclusion criteria, she met with the researcher (in a location convenient for the participant, for example, at home). During the meeting, study materials and a sample container for human milk collection were provided. If necessary, the researcher answered the participant’s questions. However, if it was more convenient for the participant or due to safety reasons raised by the COVID-19 pandemic, study materials and sample container were sent to the participant using a parcel machine, and all communication between the researcher and a participant was provided electronically.

After receiving the study materials, participants were instructed to choose four consecutive days to carry out the study. Women were able to complete the study in a location convenient for them (for example, at home). Participants were asked to complete a 72-h food diary and provide at least 10 mL of pooled human milk sample within a 24-h period (milk expressed from the feeding breast after the end of nursing, expressed from various breastfeeding sessions). During the sampling process, participants were instructed to store a prelabelled propylene container with expressed human milk in the refrigerator (~4 °C). After, it was placed in the household freezer (approximately −18 °C).

Participants were also asked to complete a self-administered food frequency questionnaire (FFQ) at any convenient time during the four days of the study period. The FFQ consisted of 71 food products and drinks for which the frequency of consumption was estimated by the participants (participants had to mark how frequently specific products had been consumed one month before participating in the study). All food products and drinks in the FFQ were divided into 15 categories and median intake values for these food products and drinks categories were after evaluated. The categories were the following:starchy foods (cereals, cereal products, potatoes);meat and meat products;fish and seafood;milk and dairy products;vegetables;legumes;fruits and berries;plant-based fats (nuts, seeds, vegetable oils, avocado);condiments (sauces, etc.);sweets and bakery goods;salty snacks and fast food;soft drinks;caffeine-containing drinks (coffee, black or green tea, etc.);herbal teas;alcohol.

No dietary restrictions were applied, and participants were encouraged to consume their regular diet. Volume measures (handful, teaspoon, etc.) could be used to help complete the 72-h food diary.

Demographic characteristics such as maternal age; body mass index; parity; time postpartum; child sex, birth weight and length; and breastfeeding pattern (exclusive or partial) were collected using a self-administered questionnaire undertaken during the four-day study period.

A second visit was arranged after participants had completed the required tasks. During a second meeting (on-site), the researcher collected all the study materials (the completed 72-h diary, FFQ, etc.) and a human milk sample from each participant. A bag with ice packs was used to transport frozen human milk samples to the laboratory’s freezer where they were stored until the analysis (−18 ± 3 °C).

The current Latvian Food composition database lacks dietary data regarding the fatty acid composition of different food products. Therefore, the Finnish Food Composition Database Fineli [15] was used to calculate energy value, protein, carbohydrates, fibre, sugars, fat and fatty acids, including *trans* fatty acids, intake among the study participants. If a participant was using dietary supplements during the study period, nutritional information was sourced from the manufacturers’ website and included in nutrient calculations.

### 2.2. Analysis of Human Milk Samples

All samples were analysed within two months of arrival at the laboratory. Before the analysis, frozen samples were thawed in warm water (~55 °C) and homogenised using Vortex (VXMNDG Vortex Mini digital, OHAUS Corporation, Parsippany, NJ, USA).

An in-house method BIOR-T-012-131-2011 (developed based on standards ISO 12966-1:2014, ISO 12966-2:2017 and ISO 12966-4:2015) was used to determine the following *trans* fatty acids in human milk samples:elaidic acid (C18:1 n9*t*);vaccenic acid (C18:1 n11*t*);linolelaidic acid (C18:2 n9*t*, n12*t*);rumenic acid (C18:2 n9*c*, n11*t*).

Samples (10.00 g) were weighted in 50-millilitre polypropylene test tubes with screw caps. Fats were extracted with 20 mL of hexane/acetone (1:1) solution. After, the mixture was mixed for 30 min using a rotator (Multi RS-60 Biosan, Rīga, Latvia) and centrifuged (3500 rpm, 10 min) (CM-6MT, Elmi, Newbury Park, CA, USA) to separate the layers of the solution. Approximately 10 mL of the upper layer of the solution was transferred into a 15-millilitre propylene screw cup test tube and evaporated with nitrogen (45 °C). Then, 100 μL of concentrated residue was transported into a 22-millilitre glass tube and dissolved in 2 mL of isooctane. To this, 200 μL of sodium methoxide was added, and the solution was mixed twice (for ~20 s each time, with 30 to 60 s pause between) using a vortex mixer (Maxi Mix II Thermo Fisher Scientific, Waltham, MA, USA). The solution was then rested for 1 to 2 min, and after, 2 mL of sodium chloride solution (40%) was added. After, the solution was mixed for one more time (20 to 30 s using a vortex mixer) and centrifuged (2500 rpm, 5 min). In the 250-microlitre autosampler container, 150 μL of isooctane and 50 μL of the organic layer were transported. Then, 20 μL of the aliquot was transferred into a 1.5-millilitre glass vial and evaporated with nitrogen (45 °C). The dry residue was reconstructed with 100 μL of hexane.

The Discovery^®^ Ag-Ion SPE column, 750 mg/6 mL (Sigma Aldrich Company, St. Louis, MO, USA) was conditioned with 4 mL of acetone and equilibrated with 4 mL of hexane. The 100 μL of prepared fatty acid methyl ester extract in hexane were loaded on the column. Fraction one was eluated with 6 mL of hexane:acetone (96:4) and eluant collected using a 15-millilitre propylene tube. This fraction contained saturated fatty acids and *trans* monounsaturated fatty acids. Fraction two was eluated with 4 mL of hexane:acetone (90:10) and eluant collected using a 15-millilitre propylene tube. This fraction contained *cis* monounsaturated fatty acids and *trans* polyunsaturated fatty acids. Fraction three was eluated with 4 mL of acetone (100%) and eluent was collected using a 15-millilitre propylene tube. This fraction contained all other polyunsaturated fatty acids. Each fraction was evaporated with nitrogen over a water bath (Biosan, Latvia) (40 °C). The dry residue was reconstituted in 80 mL of hexane for the final step—gas chromatography analysis.

Fatty acid analysis was conducted using gas chromatography with flame ionisation detection (Agilent 6890N, Agilent Technologies Incorporated, Santa Clara, CA, USA). Aliquots of the fatty acid methyl esters (1 μL) were injected at a 50:1 split ratio and at 260 °C into a Restek’s Rt-2560 column (100 m × 0.25 mm × 0.20 μm) (Restek Corporation, Bellefonte, PA, USA). The flow rate of the helium carrier gas was 1.0 mL min^−1^. The initial oven temperature of 100 °C was held for two minutes, then increased to 180 °C at 4 °C min^−1^, 210 °C at 2 °C min^−1^, 223 °C at 0.8 °C min^−1^ and 240 °C at 20 °C min^−1^ and held for another ten minutes (total time 64.1 min). The detector temperature was 250 °C, H_2_ flow—40 mL min^−1^, airflow—450 mL min^−1^.

Using authentic standards (Supelco 37 component FAME mix, Supelco Incorporated, Germany and additionally vaccenic acid and rumenic acid standards from Sigma Aldrich Company, St. Louis, MO, USA), the fatty acid methyl esters release times were determined. After, the fatty acid methyl esters chromatographic peaks were identified in the analysis of human milk samples. All identified fatty acid methyl esters peak areas were determined, expressing them as a percentage of the peak area of the sum of all peak areas (%, in total 38 fatty acids were analysed). The mean value of two parallel repetitions of the analysis was accepted as the final result.

### 2.3. Statistical Analysis of the Data

All data were compiled using Microsoft Excel 2019 and after, International Business Machines (IBM) Statistical Package for the Social Sciences (SPSS), version 23, was used for data statistical analysis. All values were expressed as median ± interquartile range and minimal–maximal values. Spearman’s rank order correlation (ρ) was used to measure correlations between the *trans* fatty acid level in human milk and maternal dietary habits. A *p*-value of ≤0.05 was considered statistically significant.

## 3. Results

### 3.1. Characteristics of the Participants

The characteristics of the participants are compiled in Table 1 and Table 2. According to body mass index calculations, 55 participants had a normal body weight, 12 participants were overweight and 3 participants were obese. Half of the participants (46%) were primiparas. Almost all the participants (97%) had a child younger than one year old.

### 3.2. Dietary Habits and Nutrient Intakes among the Participants

The evaluation of FFQ revealed that important food product groups such as starchy foods, vegetables, etc. were not consumed on a daily basis (Figure 1).

For example, only 27% of the participants consumed starchy foods such as cereals, cereal products and potatoes daily. Meat & meat products were consumed every other day by 29% of the participants. Around 30% of the participants consumed milk and dairy products every or every other day. Approximately 20% of the participants completely avoided milk and dairy products during the lactation period. The main reason for this was an infant having a cow’s milk protein allergy (10% of the participants). Other reasons were vegetarian or vegan dietary patterns (4% of the participants). The remaining participants (6%) did not state why milk and dairy products were avoided.

Fish and seafood was avoided (17% of the participants) or consumed rarely (57% of the participants). Vegetables and legumes were consumed on a daily basis by 6 and 10% of the participants, respectively. Fruits and berries were consumed every other day by 13% of the participants, but plant-based fats (nuts, seeds, vegetable oils, etc.) were consumed on a daily basis by 19% of the participants. Condiments were mostly consumed seldomly (41% of the participants). Half of the participants consumed sweets and baked goods seldomly and 60% consumed salty snacks and fast foods seldomly. Most of the participants (56%) avoided soft drinks, but caffeine-containing drinks or herbal teas were mostly consumed on a daily basis (26 and 41% of the participants, respectively). Most of the participants (79%) reported an avoidance of alcohol during the lactation period.

The median energy and nutrient intakes calculated from the 72-hour food diaries are summarised in Table 3.

If compared to nutritional guidelines (Table 3), the participants were consuming less energy and carbohydrates than recommended. Additionally, fibre intake was a little lower than recommended. Sugar intake compiled around 16% of total energy intake, and fat intake was higher than recommended. Protein intake was within nutritional guidelines.

Saturated fatty acid consumption exceeded 10% of total energy intake. Monounsaturated and polyunsaturated fatty acid intake was within recommendations. Linoleic acid consumption exceeded 4% of total energy intake. α-linolenic acid and total *n*-3 polyunsaturated fatty acid consumption reached the recommended intake of at least 0.5 and 1% of total energy, respectively. Median daily docosahexaenoic acid intake reached only 50% of the recommended daily intake.

No specific values have been set regarding recommended *trans* fatty acid intake during lactation, but both Nordic Nutrition guidelines [17] and European Food Safety Authority scientific opinion [18] suggest that the *trans* fatty acid intake during lactation should be as low as possible.

Maternal *trans* fatty acid intake was significantly higher among those with higher intakes of milk and dairy products (ρ = 0.372, *p* = 0.003) and sweets and bakery goods (ρ = 0.305, *p* = 0.017). Opposingly a significantly lower intake of *trans* fatty acids was noted among the participants with a higher intake of vegetables (ρ = −0.336, *p* = 0.008) and legumes (ρ = −0.262, *p* = 0.041).

### 3.3. Trans Fatty Acid Level in Human Milk

The median ± interquartile range, as well as the minimal–maximal values of the *trans* fatty acids level in human milk samples are summarised in Table 4.

Vaccenic acid was the most predominant *trans* fatty acid found in human milk, compiling approximately 70% of total *trans* fatty acids. There was a significant positive correlation found between elaidic acid and rumenic acid (ρ = 0.331, *p* = 0.009) as well as linolelaidic acid and rumenic acid (ρ = 0.462, *p* < 0.0005) level in human milk.

### 3.4. Trans Fatty Acid Level in Human Milk and Its Relation to Maternal Dietary Habits

A strong positive correlation was found for milk and dairy product intake and elaidic acid level in human milk (ρ = 0.632, *p* < 0.0005). Vaccenic acid level was associated with meat and meat products (ρ = 0.348, *p* = 0.006) and milk and dairy products intake (ρ = 0.277, *p* = 0.031). Total *trans* fatty acid level was related to the habitual intake of meat and meat products (ρ = 0.296, *p* = 0.021) and milk and dairy products (ρ = 0.566, *p* < 0.0005). There was a moderate direct association between maternal *trans* fatty acid intake and total *trans* fatty acid level in human milk (ρ = 0.341, *p* = 0.007).

## 4. Discussion

The content of *trans* fatty acids in natural products and, therefore, the intake of naturally occurring *trans* fatty acids is relatively constant [4,13]. On the other hand, the intake of industrially produced *trans* fatty acids within the European Union has been gradually decreasing due to the regulation rules setting the maximum permitted amount of *trans* fatty acids in food products [13,14,19]. Due to the new regulations, industrially produced food products have been reformulated and ingredients such as partially hydrogenated fats replaced, mainly with palm oil [13].

Additionally, this study shows a low *trans* fatty acid intake among lactating women (Table 5). This could be due to the above-mentioned changes in European legislation, which, therefore, now results in a lower total intake of *trans* fatty acids among lactating women in Europe.

Overall, the *trans* fatty acid intake among lactating women in Europe has not been comprehensively analysed in the past decade. Few studies that have been conducted differ in methodology and have low participant numbers (<100 participants) (Table 5). Therefore, data can only be compared in the context of the significant reduction in the *trans* fatty acid content of the food due to legislation changes and, therefore, result in the lower consumption of *trans* fatty acids among lactating women in Europe.

According to the obtained correlations, the *trans* fatty acid intake in this study was related to both milk and dairy products and sweet and bakery goods intake. Researchers from Croatia [21] also found results, identifying that the dominant dietary sources of *trans* fatty acids among lactating women were sweets and bakery goods and milk and dairy products.

Not only *trans* fatty acid intake, but also *trans* fatty acid composition among lactating women in Europe has not been comprehensively analysed in recent years (Table 6). Therefore, it is not possible to compare the data. However, overall, the total *trans* fatty acid level in human milk among lactating women from Europe is lower than the data reported from the United States (Table 6). Referring to the results from this study and taking into account the recent legislation changes in the European Union, it can be speculated that the *trans* fatty acid intake as well as the *trans* fatty acid levels in human milk among lactating women in Europe has likely decreased in the past few years.

Previous studies have indicated that vaccenic acid can be converted to the rumenic acid in the mammary glands during lactation [5,12]. Nevertheless, the conversion rate is low, with less than 10% of the rumenic acid coming from the endogenous synthesis of vaccenic acid [5]. This could explain why no significant correlation between the vaccenic acid and rumenic acid levels in human milk was found in this study.

Vaccenic acid is the predominant *trans* fatty acid found in animal origin fat [4]. Therefore, the strong positive association in this study between vaccenic acid levels and the intake of meat and meat products and milk and dairy products was not surprising. Although, it should be noted that the overall milk and dairy products and meat and meat products intake among the participants was not high, with the majority of participants consuming these food products only a few times a week. Approximately 20% of participants completely avoided milk and dairy products during the lactation period (Figure 1). This was predominately due to infants having a cow’s milk protein allergy.

Precht and Molkentin (1999) [23] have also observed a higher vaccenic acid level in human milk among German lactating women with a higher consumption of milk and dairy products. A study from the Netherlands [22] suggests that a maternal diet rich in meat and dairy products (especially organically produced) is associated with a lower elaidic acid/vaccenic acid ratio in human milk (ratio of 1.27 for conventional diets versus the ratio of 0.86 for a diet rich in organically produced meat and dairy products). Although milk and dairy products, as well as meat and meat products, intake was not high, this study reports a lower elaidic/vaccenic acid ratio in human milk of 0.29 compared to the study from the Netherlands [22].

Although a higher elaidic acid intake and, therefore, a higher elaidic acid level in human milk is usually associated with industrially produced food products such as sweets and bakery goods intake [4,25], in this study, a higher elaidic acid level in human milk was associated with milk and dairy product intake. This obtained correlation could be explained by the fact, that cow’s milk can also contain traces of elaidic acid, and its content can increase with the heat treatment of milk [26].

A higher rumenic acid level in human milk (0.34%) has been reported among lactating women who are predominantly consuming meat and dairy products of organic origin compared to women on conventional diets (0.25%) [27]. Data were not collected to determine if participants consumed conventional or organically produced ruminant products in this study, but we speculate that participants were mostly consuming conventionally produced meat and meat products and milk and dairy products. Additionally, within this study, the FFQ results suggested that women less frequently consumed milk and dairy products as well as meat and meat products. This could potentially explain why lower rumenic acid levels are reported when compared to data from other countries (Table 6).

Similar to observations reported from other countries [10,21,23], this study also reports that the *trans* fatty acid level in human milk is directly linked to the maternal dietary intake of *trans* fatty acids. However, it seems that the *trans* fatty acid level in human milk among lactating women in Latvia is more closely related to the intake of ruminant fat (milk and dairy products and meat and meat products). This differs from previous studies that have found that *trans* fatty acid levels in human milk were related to the intake of sweets and bakery items in Polish women [11] and sweets, bakery goods, fried food, dairy products, margarine and sausages in Croatian women [21]. Other studies [28] have not observed a direct link between the maternal intake of *trans* fatty acids during lactation and their level in human milk.

Obtained results indicate that maternal dietary habits during lactation are an important factor influencing the quality of the nutrition received by the infant via human milk. Women from Latvia during the lactation period are predominantly consuming *trans* fatty acids of natural sources (ruminant food products). Thanks to the national legislation rules in Latvia [14], women are able to choose industrially produced foodstuff without a high amount of *trans* fatty acids, which can have an adverse effect both on the mother and the infant.

Nevertheless, there are currently no nutritional guidelines in Latvia for women during their lactation period. To increase awareness of women’s nutritional needs during the lactation period, the national authorities of Latvia could adopt dietary guidelines that encourage women to consume more milk and dairy products (preferably organically produced), to increase the conjugated linoleic acid level in human milk, potentially providing further health benefits to the breastfed infant.

## 5. Conclusions

The dietary data suggest that the *trans* fatty acid intake among lactating women in Latvia is generally low and does not raise concern. This is probably due to the regulation rules in the Europe Union setting the maximum permitted amount of *trans* fatty acids in food products. Vaccenic and elaidic acid were the most predominant *trans* fatty acids in human milk, which was largely determined by the consumption of milk and dairy products.

## Figures and Tables

**Figure 1 nutrients-13-02967-f001:**
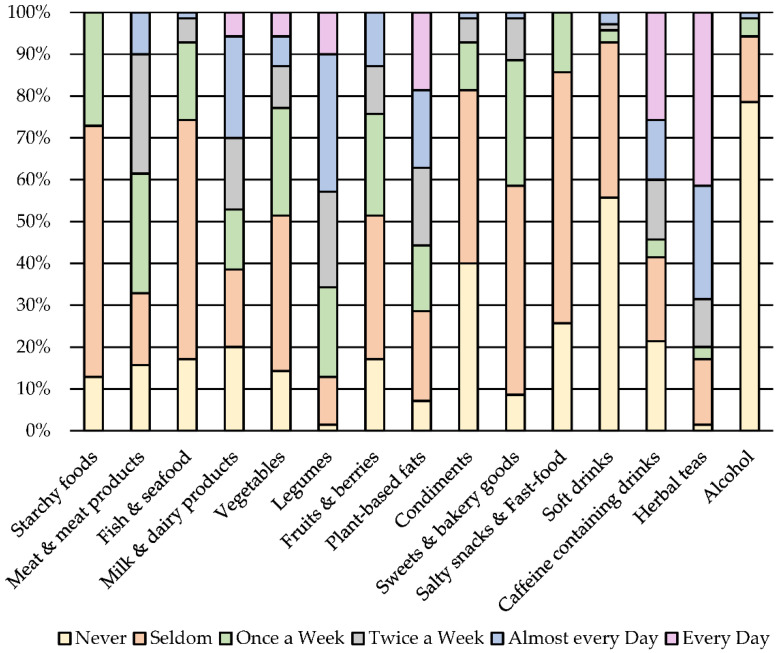
Median intake frequency of food products and drinks one month before participating in the study.

**Table 1 nutrients-13-02967-t001:** Maternal characteristics.

Characteristics	Median ± Interquartile Range (Minimal–Maximal Values)
Age (years)	31 ± 7 (23–45)
Maternal body mass index ^1^	22.28 ± 3.79 (18.51–36.57)
Parity	2 ± 1 (1–5)
Breastfeeding pattern	48—exclusive breastfeeding, 22—partial breastfeeding

^1^ calculation based on weight and height values given by the participant. No anthropometrical measures were performed during this study.

**Table 2 nutrients-13-02967-t002:** Child’s characteristics.

Characteristics	Median ± Interquartile Range (Minimal–Maximal Values)
Age (months)	3 ± 4 (1–27)
Sex	34—females, 36—males
Birth weight (kg)	3.61 ± 0.64 (1.63–5.50)
Birth length (cm)	54 ± 3 (42–61)

**Table 3 nutrients-13-02967-t003:** Daily energy and nutrient intakes in comparison to nutritional guidelines (median values from 72-h food diary).

Energy Value or Nutrient (Unit)	Median ± Interquartile Range (Minimal–Maximal Values)	Nutritional Guidelines
Energy value (kcal)	2007.42 ± 498.31 (827.09–3191.99)	2340–3110 kcal [16]
Protein (E%)	15.52 ± 5.80 (8.01–43.08)	10–20 E% [16,17]
Carbohydrates (E%)	38.81 ± 8.38 (11.90–52.79)	45–60 E% [16,17]
Sugars (E%)	16.15 ± 6.85 (0.93–27.94)	≤10 E% (free sugars) [16,17]
Fibre (g)	22.72 ± 12.01 (7.05–48.84)	at least 25–35 g [17]
Fat, total (E%)	42.69 ± 8.63 (29.78–56.79)	25–30 E% [16] 25–35 E% [18] 25–40 E% [17]
Saturated fatty acids (E%)	14.09 ± 6.62 (5.45–20.77)	≤10 E% [16,17] As low as possible [18]
Monounsaturated fatty acids (E%)	15.61 ± 4.29 (9.32–27.79)	10–20 E% [17]
Linoleic acid (C18:2, *n*-6) (E%)	5.40 ± 2.95 (2.37–17.46)	4 E% [18]
α-linolenic acid (C18:3, *n*-3) (E%)	1.07 ± 0.71 (0.33–3.16)	at least 0.5 E% [17,18]
Eicosapentaenoic acid (C20:5, *n*-3) (mg)	18.01 ± 255.35 (0.00–1222.69)	no guidelines
Docosahexaenoic acid (C22:6, *n*-3) (mg)	117.87 ± 234.39 (0.00–3369.78)	200 mg [17]
*n*-3 polyunsaturated fatty acids (E%)	1.20 ± 0.95 (0.11–5.51)	at least 1 E% [17]
*n*-6 polyunsaturated fatty acids (g)	12.44 ± 6.74 (2.93–34.06)	no guidelines
Polyunsaturated fatty acids (E%)	7.00 ± 3.53 (3.73–17.92)	5–10 E% [16,17]
*Trans* fatty acids (g)	0.54 ± 0.79 (0.00–1.82)	as low as possible [17,18]

**Table 4 nutrients-13-02967-t004:** *Trans* fatty acid level (% of total fatty acids) in human milk.

*Trans* Fatty Acids	Median ± Interquartile Range (Minimal–Maximal Values)
Elaidic acid (C18:1 n9*t*)	0.50 ± 0.40 (0.10–1.40)
Vaccenic acid (C18:1 n11*t*)	1.70 ± 0.50 (0.90–2.20)
Linolelaidic acid (C18:2 n9*t*, n12*t*)	0.10 ± 0.10 (<0.10–0.30)
Rumenic acid (C18:2 n9*c*, n11*t*)	0.10 ± 0.10 (<0.10–0.30)
*Trans* fatty acid, total ^1^	2.30 ± 0.60 (1.00–3.20)

^1^ Rumenic acid is not included in the sum of total *trans* fatty acids [19].

**Table 5 nutrients-13-02967-t005:** *Trans* fatty acid intake (g per day) among lactating women. Data from different countries.

Latvia (*n* = 70)	Romania (*n* = 33) [20]	Croatia (*n* = 83) [21]	Poland (*n* = 69) [11]
0.54 ± 0.79 (0.00–1.82)	0.95 (0.81–1.07)	2.00 ± 0.90	5.76 ± 2.77

**Table 6 nutrients-13-02967-t006:** The *trans* fatty acid level (% of total fatty acids) in human milk. Data from different countries.

*Trans* Fatty Acids	Latvia (*n* = 70)	The Netherlands (*n* = 186, Conventional Diet Group) [22]	Germany (*n* = 40) [23]	Greece (*n* = 127) [24]	The United States (*n* = 81) [25]
Elaidic acid	0.50 ± 0.40	0.61 ± 0.27	0.34 ± 0.14	0.57 ± 0.38	0.65 ± 0.25
Vaccenic acid	1.70 ± 0.50	0.48 ± 0.21	0.68 ± 24	No data	0.04 ± 0.01
Linolelaidic acid	0.10 ± 0.10	No data	0.12 ± 0.08	0.11 ± 0.17	0.91 ± 0.28
Rumenic acid	0.10 ± 0.10	0.25 ± 0.07	0.40 ± 0.09	No data	0.43 ± 0.10
*Trans* fatty acids, total ^1,2^	2.30 ± 0.60	3.26 ± 1.06	3.81 ± 0.97	0.68 ± 0.44	7.00 ± 2.30

^1^ Rumenic acid is not included in the sum of total *trans* fatty acids [19]. ^2^ Not all individual *trans* fatty acids are reported in the table; therefore, the total *trans* fatty acid level may be higher than the total sum of individual *trans* fatty acids reported in the table.

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
