# Peer review of "Trans Fatty Acids in Human Milk in Latvia: Association with Dietary Habits during the Lactation Period"

_nutrients, 2021, doi:10.3390/nu13092967_

Round 1

Reviewer 1 Report

This is an interesting, well-conceived, and clearly written manuscript on an important topic to understand the metabolic relationships and reciprocal influences between mother and infant during lactation. A good point is its appropriate methodologies of fatty acid determination via gas chromatography with flame ionization detection after transesterification.
Say that, please 
1-    Improve the introduction including also other components in human milk with a focus on the microRNAs because they fluctuate as the fatty acids in breast milk
2-    adds the chemical structure of the fatty acid tested in the 2.2 section.
3-    try to explain how trans-fatty acids from the diet can cross the intestinal membrane to go into human milk in the discussion section.

Author Response

We would like to thank the reviewer for the detailed comments and suggestions provided for the improvement of the manuscript (nutrients-1345236). We believe that the comments have identified important areas which required improvement. After completion of the suggested edits, the revised manuscript has benefited from an improvement in the overall presentation and clarity. Below, you will find a point-by-point description of how each comment was addressed in the manuscript. Original comments in boldface, responses in regular typeface.

1-improve the introduction including also other components in human milk with a focus on the microRNAs because they fluctuate as the fatty acids in breast milk.

We agree that microRNAs in human milk is also an important topic to discuss. However, the main focus of this article was about trans fatty acids in human milk. Therefore, we decided not to include in the introduction part section about microRNAs. But during her PhD studies, the researcher Līva Aumeistere have analyzed also other components of the human milk (fat, protein, lactose, other fatty acids, essential and potentially toxic elements), and Līva Aumeistere is planning to publish another article about overall variability of human milk composition, where the section about microRNAs will be also included.

2-add the chemical structure of the fatty acid tested in the 2.2 section. 

We decided not to add in the 2.2. section the chemical structure of the analyzed fatty acids. We believe it will not provide any additional information for the article. Also, structural chemical formula is the format how the information is usually provided in this type of articles.

3-try to explain how trans fatty acids from the diet can cross the intestinal membrane to go into human milk in the discussion section.

We decided to corporate this information in the Introduction part (Lines 33 to 39).

Reviewer 2 Report

General remark: 
A clear distinction should be made between the trans fatty acids (TFA, Trans fat) and CLA isomers (Conjugated linoleic acid). According to the definition set out in Annex I of Regulation (EC) No 1169/2011 of the European Parliament and of the Council, „trans fat means fatty acids with at least one non-conjugated (namely interrupted by at least one methylene group) carbon-carbon double bond in the trans configuration”. The cited definition does not include CLA, including rumenic acid (18: 2 c9, t11). Therefore, the CLA cannot be counted towards the sum of the TFA. 
Other comments: 
- Introduction page 2, lines 55 – 56:  This statement should be corrected. 
In 2019, Commission Regulation (EU) 2019/649 amending Annex III to Regulation (EC) No 1925/2006 of the European Parliament and of the Council as regards trans fat, other than trans fat naturally occurring in fat of animal origin has entered into force. According to this regulation the content of trans fat, other than trans fat naturally occurring in fat of animal origin, in food intended for the final consumer and food intended for supply to retail, shall not exceed 2 grams per 100 grams of fat. Food which does not comply with this Regulation may continue to be placed on the market until 1 April 2021. 

- Materials and Metod  
2.1 Collection of Human milk samples  More detailed data on the collection of human milk samples should be provided. Were the samples  stored  before  the  analysis?  and  for  how  long  and  under  what  conditions (temperature)? 
2.2 Analysis of Human Milk Samples 
Describe how the milk samples were prepared for testing (thawed ?, mixed? sonicated?). More details on the chromatographic analysis should be provided, e.g. oven temperature, analysis time or quote the official method. 

Results 
Table 3 should be corrected. CLA (18: 2 cis 9, trans 11) was included in the sum of the trans fatty acids. Comment as above. 

Discussion  
I would like to point out that the content of trans fatty acids in natural products (r-TFA) is practically constant, only slight fluctuations are observed. On the other hand, the amounts of industrially produced trans fats (i-TFA) in the last 10 - 15 years in most countries were gradually decreasing. It was related to the reformulation of products and the replacement of partially  hydrogenated  fats  mainly  with  palm oil.  These  activities were  related to  the introduction of the obligation to label food with the content of TFA, among others in the USA and Canada or the introduction of maximum permissible levels, e.g. Denmark, Austria and, more recently, the entire EU. Therefore, a comparison of TFA consumption in different countries should be based on similar periods. In Table 4, the authors compared their own results to the data from 2013 (Croatia) and 2018 (Romania) and 2003 (Poland). The latter are 10-15 years away from the previous ones. Therefore, these are incomparable data. They can only be compared in the context of the significant reduction in the TFA content of the food and the resulting lower current  consumption.

Author Response

We would like to thank the reviewer for the detailed comments and suggestions provided for the improvement of the manuscript (nutrients-1345236). We believe that the comments have identified important areas which required improvement. After completion of the suggested edits, the revised manuscript has benefited from an improvement in the overall presentation and clarity. Below, you will find a point-by-point description of how each comment was addressed in the manuscript.  Original comments in boldface, responses in regular typeface.

A clear distinction should be made between the trans fatty acids (TFA, Trans fat) and CLA isomers (Conjugated linoleic acid). According to the definition set out in Annex I of Regulation (EC) No 1169/2011 of the European Parliament and of the Council, „trans fat means fatty acids with at least one non-conjugated (namely interrupted by at least one methylene group) carbon-carbon double bond in the trans configuration”. The cited definition does not include CLA, including rumenic acid (18: 2 c9, t11). Therefore, the CLA cannot be counted towards the sum of the TFA.

We than the Reviewer for this important notice. We have corrected this mistake. Recalculations regarding total trans fatty acid level in human milk were made (excluding rumenic acid from the total sum of fatty acids).

Other comments:

- Introduction page 2, lines 55 – 56:  This statement should be corrected.

In 2019, Commission Regulation (EU) 2019/649 amending Annex III to Regulation (EC) No 1925/2006 of the European Parliament and of the Council as regards trans fat, other than trans fat naturally occurring in fat of animal origin has entered into force. According to this regulation the content of trans fat, other than trans fat naturally occurring in fat of animal origin, in food intended for the final consumer and food intended for supply to retail, shall not exceed 2 grams per 100 grams of fat. Food which does not comply with this Regulation may continue to be placed on the market until 1 April 2021.

We corrected this information and added reference to the regulations from the European Parlament (Lines 70 to 73).

- Materials and Metod 

2.1 Collection of Human milk samples  More detailed data on the collection of human milk samples should be provided. Were the samples  stored  before  the  analysis?  and  for  how  long  and  under  what  conditions (temperature)?

We provided more detailed information regarding the human milk sample collection (Lines 95 to 109).

2.2 Analysis of Human Milk Samples

Describe how the milk samples were prepared for testing (thawed ?, mixed? sonicated?). More details on the chromatographic analysis should be provided, e.g. oven temperature, analysis time or quote the official method.

We provided more detailed information regarding the analysis of human milk samples (Lines 155 to 209).

Results

Table 3 should be corrected. CLA (18: 2 cis 9, trans 11) was included in the sum of the trans fatty acids. Comment as above.

Recalculations regarding total trans fatty acid level in human milk were made (excluding rumenic acid from the total sum of fatty acids) and corrections were made in the Table 3.

Discussion 

I would like to point out that the content of trans fatty acids in natural products (r-TFA) is practically constant, only slight fluctuations are observed. On the other hand, the amounts of industrially produced trans fats (i-TFA) in the last 10 - 15 years in most countries were gradually decreasing. It was related to the reformulation of products and the replacement of partially  hydrogenated  fats  mainly  with  palm oil.  These  activities were  related to  the introduction of the obligation to label food with the content of TFA, among others in the USA and Canada or the introduction of maximum permissible levels, e.g. Denmark, Austria and, more recently, the entire EU. Therefore, a comparison of TFA consumption in different countries should be based on similar periods. In Table 4, the authors compared their own results to the data from 2013 (Croatia) and 2018 (Romania) and 2003 (Poland). The latter are 10-15 years away from the previous ones. Therefore, these are incomparable data. They can only be compared in the context of the significant reduction in the TFA content of the food and the resulting lower current  consumption.

We agree with the Reviewer. We corrected the Discussion section pointing out that both dietary trans fatty acid intake as well as trans fatty acid intake among lactating women in Europe has not been comprehensively analyzed in the recent years. Therefore, data cannot be compared. We added the statement that due to changes in legislation and reformulation of food products, trans fatty acid intake have lowered in Europe, and it also results in a lower level of trans fatty acids in human milk (Corrected lines 287 to 302, 311 to 319).

Reviewer 3 Report

This paper aimed to report the trans fatty acid level in human milk and how maternal dietary habits affects the trans fatty acid composition of human milk in Latvian women. The study included 70 women who were least one month postpartum. These women were asked to supply 10mL of expressed hind milk, 72 hour food diary, and food frequency questionnaire.

Overall, this manuscript provides relevant research for the Latvian community being the first to report the trans fatty acid level in human milk and how maternal dietary habits can affect the trans fatty acid composition of human milk in Latvia. The paper provides insight into the current limitations of nutrition guidelines and possible effect this may have on infantile nutrition. Having said this there are various limitations to this study that need to be addressed, particularly surrounding the methodology, reported results, figure use, sentence structure, grammar, and reference to previous research.

In general, an extensive review of the writing style and grammar is required to increase readability. Please also consider the below specific recommendations:

Abstract

Sentence 1-3:

Good summarizing title, however, no reference that this study was conducted in Latvia. Query changing to:

“trans Fatty Acids in Human milk. Association with Dietary Habits during the Lactation Period in Latvia”

Sentence 10-11:

“mainly” does not fit the theme of this sentence. Query remove and replace with predominately.

Sentence 11-12:

This sentence structure does not make sense. Conjunctions are required to allow the sentence to flow. Query replace with:

“…. Was to determine the trans fatty acid level in human milk among lactating women from Latvia, and to evaluate how maternal dietary habits affect the trans fatty acid composition of human milk.

Sentence 16-17:

As above. Query replace with:

“Overall, the dietary intake of trans fatty acids ….”

Sentence 17-18:

As above. Query replace with:

The Total trans fatty acid level in the human milk samples was 2.40 % ± 0.60 %.”

Sentence 18-19:

Poor sentence structure and lack of proper English. Query replace with:

“This study reports the highest vaccenic acid and lowest rumenic acid levels recorded in human milk samples when compared to previous studies (references)”

Sentence 19-20:

This sentence structure does not make sense. Query replace with:

“The composition of trans fatty acids found in human milk was associated with maternal dietary habits.”

Sentence 20-21:

Additional words used that are not required.  Remove “the”

A higher elaidic acid, vaccenic acid and total trans fatty acid level in human milk was found among the participants with a higher milk & dairy product intake.

Sentence 22-23:

Incorrect grammar. As two measures are reported level needs to be plural e.g. levels.

“Meat & meat product intake was associated with a higher vaccenic acid and total trans fatty acid levels in human milk”

Sentence 23-24:

This sentence structure does not make sense. Query replace with:

“A moderate association was also established between maternal trans fatty acid intake and the total trans fatty acid level in human milk.”

Introduction

General: 

Further introduction and evidence are required surrounding the relevance/benefits of trans fatty acids in breastmilk. This is not currently discussed.

Sentence 29-31:

first half of year” is overly wordy. Please rewrite e.g.,human milk is the only nutrient source during this time” – you have already referenced that you are talking about this first six months of life.

Please also correct grammar surrounding percentages e.g 15-20%

Sentence 31-32

Sentence can be refined e.g.“Nutrients for human milk are derived from maternal body stores and those absorbed directly from the maternal diet”

Sentence 32-33

This sentence structure does not make sense. Conjunctions are required to allow the sentence to flow.

Query replace with:

“Therefore, a well-balanced diet during lactation is critical both for the mother and breastfed infant”.

Additionally, the word critical suggests that serious consequences will occur if the diet is not optimal. Query replace with a word that does not have a causative relationship e.g. important.

Sentence 35-37

This sentence has a double negative “already previously”. Please select one e.g., “previously reported”

This sentences grammar can also be improved. Query replace with:

“For example, we have already previously reported that fish intake reflects on the docosahexaenoic acid level in human milk among lactating women from Latvia.”

Sentence 47

Incorrect grammar. “the’ not required before health

Sentence 47-48

Sentence flow can be improved. Query replace “The” with “This”.

Sentence 49-50

Poor word choice. “On the opposite” is an odd choice of words. Query replace with: opposingly.

Grammatical errors with poor sentence flow. Query replace with:

On the opposite Opposingly, conjugated linoleic acid is associated with health related benefits, for example such as an antiatherogenic effect for XXX [7]”

Who is an antiatherogenic affect beneficial for? Mum or baby or both? Are there any other specific benefits of trans fatty acids in breastmilk?

Sentence 47-48

Grammatical errors with poor sentence flow. “The” required to start the sentence. Query move “currently” (and change to current) before general position e.g. “The current general position currently is that human body…”

Sentence 67-68

Sentence flow can be improved. Query replace with:

“As breastmilk is the sole source of nutrition for 15-20% of infants in the first six months of life, the quality of human milk is of great importance.”

Sentence 68-71

Grammatical errors. Conjunctions are required to allow the sentence to flow. Query add “the” after affects e.g. “…..affects the trans fatty acid composition….”

Materials and methods

General:

How many study appointments occurred?

Where was the research conducted? E.g., where were the research facilities or were home visits conducted?

Sentence 73-74

Did other women sign up for the study?

Did anyone drop out of the study?

Grammatical errors. Conjunctions are required to allow the sentence to flow.

e.g. “…..during the period between January and till December year  2020.

Sentence 74

Were there any exclusion criteria?

Sentence 78

How many months (maximum) postpartum were women eligible to take part? Did postpartum age affect the results in any way?

Sentence 86-87

Poor word choices and sentence structure. Query replace with:

During the study, participants had were asked to complete a self-administered 72-hour food diary and provide a and to obtain a 10mL pooled human breastmilk sample (hindmilk expressed from various breastfeeding sessions) of ~10 ml within next within 24 hours of their first appointment (unsure if this is correct or not??). They were also asked to complete a food frequency questionnaire (FFQ).

When did each part of the study take part?

How was hind milk determined? More information required here.

Note food frequency questionnaire can be abbreviated to FFQ for further references. Please change in remaining text.

Sentence 88

Remove sentence – add self-administered to sentence 86-87 as above.

Sentence 88-89

The same thing has been said twice. Please refine sentence e.g.

“ No dietary restrictions were applied and participants were encouraged to consume their regular diet., participants were able to consume ad libitum diet.”

Sentence 89-90

Sentence flow can be improved. Query replace with:

“Volume measures (handful, teaspoon etc.) were provided to participants to could be used to help complete the food diary.”

Sentence 90-94

Reshuffle sentence, add conjunctions, and remove the word “Unfortunately,” as it provides unnecessary judgment. Query replace with:

Unfortunately, The current Latvian food composition database lacks dietary data regarding on the fatty acid composition of different foodstuff. Therefore, the Finnish Food Composition Database Fineli [12] was used to calculate participants energy value, and nutrients intake, including trans fatty acids among the study participants.”

What nutrients were calculated?

Sentence 94-96

Sentence can be refined to improve readability. Query replace with:

“If participants were using dietary supplements were used during the study period, nutritional information was sourced from of them was taken from the manufacturers’ website and included in nutrient calculations the calculations of total nutrient intake.”.

Sentence 97

When did they complete the FFQ?

More information is required about the timeline of your methods.

Sentence 91-99

Sentence conjunctions required. Start sentence with “The”

Sentence 99-100

Unsure of what this sentence is saying? What categories were chosen and why?

What do you mean by median intake values?

Sentence 101-103

How was this information collected?

When was it collected?

Query would demographics be a better word than characteristics?

Sentence 105-106

Sentence conjunctions required. Query add “the” before “trans fatty acid …”

What solvents were used for extraction?

Query reword sentence – currently difficult to decipher if solvent or ”transesterified” occurred first.

Sentence 106-111

Very long sentence. Please refine.

Missing references for standards.

Sentence 111-116

Incomplete sentence. “The” required to start sentence.

Sentence 117-119

Grammatical errors.

Change “was” to “were”

How was the relative proportion of each trans fatty acid was determined?

Sentence 122-123

“The” not required before MS excel

MS abbreviation has not been described. Please write in full.

Sentence 124-126

Grammatical errors.

Add “the” before trans fatty acid

Results 129-131

Grammatical errors and the format between BMI groups differs.

“the” not required before body mass index.

Write BMI groups in the same format e.g.,” (n=55)” or “12 participants”

Remove word “but” unsure what you are trying to say. Places judgment on these participants.

Sentence 131-132

Query report percentage rather than number of participants.

Sentence 132-133

Almost is used to start two sentences in a row.

Query report percentage rather than number of participants.

Sentence 134-136 (table 1)

Table needs to be across one page. Table is hard to read. Query add lines between topics.

Sentence 137-139 (figure 1)

Query improve figure title. Is “among the participants” required? Who else would it be? What was the time frame of data collected from?

This figure provides various information that is important to the study, however, the figure is hard to read. Can you show this information in a different way that is easier to read?

If a key is being used, please refer to this as a key.

Does the table show the diet per day or across a month?

Sentence 138-139

What are important food products?

These have not been introduced in the introduction. If reporting in results this needs to be included in the introduction.

Sentence 140-142

Poor sentence structure with both past and present tenses are used. I am not sure what you are trying to say ?dairy was consumed more than cereals?

Sentence can be refined to improve readability.

Is rarely the best word choice? Would it be more accurate to compare food groups according to frequencies that were questioned?

For example, XX% of women consumed starchy foods like cereals, cereal products, and potatoes daily. were consumed 140 rarely (~60 % of the participants). Opposingly XX% of women consumed milk and milk products daily. but milk & dairy products – almost every day by only one quarter of the participants.”

Where are cereals and grains on the figure? Is this starchy foods?

Sentence 142-144

Grammatical errors. Sentence can be refined to improve readability.

Query replace with:

“Approximately 20% of the participants were completely avoiding avoided milk & dairy products during the lactation period. The main reason for this was due to mostly noting the reason – infant cow’s milk protein allergies for the infant.

What percentage of avoidance was due to allergies?

Sentence 146-147

Remove “-“ to improve readability replace with “were”.

As above is rarely the best word choice?

Sentence147-149 and 149-151

Why are the reported statistics approximately? Can you record an accurate statistic?

Sentence 149-151

You have already said this data is from the FFQ – do you need to repeat this?

Sentence 152

“the” required to start sentence.

Sentence 155-156 (table 2)

Is “among the participants” required in title?

What guidelines were used? ?references

The table is hard to read. Query add lines between topics to increase readability.

Can energetic value be simply written as energy?

Why are 2-3 guidelines reported (even when the same numbers)? E.g., total fat

Sentence 156 -176

Gap required between table and next paragraph

This whole paragraph needs to be reordered. It would make more sense to report results in groups e.g. not meeting energy carbohydrate and fibre requirements then higher amounts recorded for sugar and fat. Lastly meeting protein requirements.

Sentence 157-159

This seems to be two sentences in one. Please split.

Grammar and word choice can be improved, as per comment above.

Sentence 160-161

Reference required.

Sentence 161-164

Grammar, word choice, and sentence structure can be improved.

Remove “but”. Poor word choice,

% should be written directly after number e.g. 4%, 1%, and 0.5%

Sentence 164-166

Remove “among participants” – who else would it be?

How low? reference to guidelines what % was met.

Combine this sentence with below sentence - you say the same thing twice.

Sentence 167-170

Sentence conjunctions required. Add “the” before trans fatty acid intake

Sentence 170-173

Sentence structure can be improved. No need for bullet points. Query replace with:

“Maternal trans fatty acid intake was significantly higher among those with higher intakes of milk & dairy products (ρ = 0.372, p = 0.003) and sweets & bakery goods (ρ = 0.305, p = 0.017).”

Sentence 174-176

“On the opposite” is an odd phrase. Query reword to “Opposingly a significantly lower intake of ....”

Sentence 178-179

Sentence conjunctions required and grammar errors. American language used. Query replace with:

The median ± interquartile range, as well as and minimal–maximal values of trans fatty acids level in participants human milk samples among study participants are summarizsed in Table 3.

Sentence 181-182

Sentence structure can be improved. Query replace with: “Vaccenic acid was the most predominant ...”

% should be written directly after number e.g. 70%

Sentence 182-183

The not required before individual.

Is human milk required in this sentence?

Sentence 183-184

You have said level three times in this sentence

Is human milk required in this sentence?

Please reword sentence.

Sentence 185-186

How were they correlated?

Sentence 188-189

Is human milk required in this sentence?

Sentence 189-191, 189-191, and 191-193

Is human milk required three times in these sentences?

Sentence 191-193

Remove “also” – informal language.

Discussion

Limitations and advantages of the study is not discussed. Query include in discussion?

Sentence 196-198

Sentence structure can be improved. Query replace with:

“This study shows a significantly lower trans fatty acid intake compared to previous research investigating lactating women”

Sentence 199

Were all of these studies conducted with the same methodology?

Doe the low participant numbers impact these studies validity?

Sentence 203-205

Sentence structure can be improved. E.g. “Research from Croatia shows similar results to this study, previously identifying that ....dietary sources...”

Sentence 206-207

Sentence structure can be improved. Add “the” before total trans fatty acid ….

Sentence 207-209

Also not required to start sentence.

Query “studies” instead of researchers to increase formality of writing. 

Sentence 209-211

As above query “studies” instead of researchers to increase formality of writing. 

Sentence 212-213 (table 5)

the” required before trans fatty acid

Sentence 214-216

Various grammar errors. Query replace with:

“Not all individual trans fatty acids are reported in the table, therefore the total trans fatty acid level may be higher than the total sum of individual trans fatty acids reported in the table.

Sentence 217-218

Sentence structure can be improved e.g. “previous studies have indicated that vaccenic acid can be converted to rumenic acid during lactation”

Sentence 218-220

Sentence structure can be improved e.g.

“Nevertheless, the conversion rate is low with less than 10% of the rumenic acid in the human coming from the endogenous synthesis of vaccenic acid”

Sentence 220-221

Sentence structure can be improved e.g.

This could explain why no significant correlations between the vaccenic acid and rumenic acid level in human milk was found.

Sentence 222-225

Sentence structure can be improved e.g.

“Vaccenic acid is the predominant trans fatty acid found in animal origin fat [4]. Therefore, the strong positive association in this study between vaccenic acid levels and the intake of meat & meat products and milk & dairy products was not surprising.”

Sentence 225-227

Grammatical errors and sentence structure can be improved. E.g.

Although it should be noted that the overall, milk & dairy product and meat & meat product intake among the participants was not high with the majority of participants consuming these foods once a week or less”

Was the amount of food consumption recorded e.g. only consumed once per week but large amounts?

Sentence 227-229

Grammatical errors and sentence structure can be improved. E.g.

“Approximately 20% of participants completely avoided milk & dairy products during lactation period. This was predominately due to infants having a cow’s milk protein allergy.”

Sentence 230-235

Grammatical errors and sentence structure can be improved. 

e.g. “Precht & Molkentin (1999) [19] have also observed higher vaccenic acid levels in human milk among German lactating women with a higher consumption of milk and dairy products. Opposingly studies from the Netherlands suggest that a maternal diet rich in meat and dairy products (especially organically produced) is associated with a lower elaidic acid/vaccenic acid ratio in human milk (ratio of 1.27 for conventional diets versus ratio of 0.86 for diet rich in organically produced meat and dairy products).  

Sentence 236

Remove “even”. Does not add anything to sentence.  Replace with “a”

Remove “- only 0.29” with “of 0.29”

Sentence 240-242

Grammatical errors and sentence structure can be improved.  E.g.

“….. contain traces of elaidic acid, with previous studies showing the elaidic acid level of human milk can increase with heat treatments”

Sentence 243-245

Grammatical errors and sentence structure can be improved. E.g.

“Higher rumenic acid levels in human milk (0.34 %) have been reported among lactating women who are predominantly consuming meat and dairy products of organic origin compared to women on conventional diets (0.25 %) [23].”

Sentence 246-248

Grammatical errors and sentence structure can be improved. E.g.

“Data was not collected to determine if participants consumed conventional or organically produced ruminant products in this study”

“but we speculate that participants were mostly consuming conventionally produced meat & meat products, milk & dairy products”

Can you say this?

Is there any data from similar cohorts in Lavita to suggest this?

Sentence 248-250 and 250-252

Grammatical errors and sentence structure can be improved. E.g.

“Within this study the FFQ results suggested that women less frequently consumed milk & dairy products as well as meat & meat products. This could potentially explain why lower rumenic acid levels are reported when compared to data from other countries”

Sentence 253-255

Grammatical errors and sentence structure can be improved. E.g.

Replace “we also report” with ‘this study also reports that the trans fatty acid level in human milk is directly linked to the maternal dietary intake of trans fatty acids.”

Sentence 255-257

Grammatical errors and sentence structure can be improved. E.g.

However, it seems that the trans fatty acid level in human milk among lactating women in Latvia is more closely related to the intake of ruminant fat intake (milk & dairy products and meat & meat products).

Sentence 257-259 and 259-262

Grammatical errors and sentence structure can be improved. E.g.

“This differs to previous studies that have found that trans fatty acid levels were related to the intake of sweets and bakery items in Polish women (reference) and sweets, bakery goods, fried food, dairy products, margarine, and sausages in Croatian women (reference).”

Sentence 262-263

Grammatical errors and sentence structure can be improved. E.g.

Query “studies” instead of researchers to increase formality of writing. 

Query change “found” to “observed a”

Add “the” before “maternal intake”

Sentence 264-265

Grammatical errors and sentence structure can be improved. E.g.

Add “an” before “important”

Sentence 266-269

Grammatical errors and sentence structure can be improved.

Remove “are mostly consuming” – replace ”predominantly consume trans fatty acids from natural food sources”

“thanks to the national legislation rules in Latvia [11], women are able to choose industrially produced foodstuff without high amount of trans fatty acids who can have adverse effect both on the mother and the infant”

Was this asked in the study or an assumption?

Sentence 270-272

Grammatical errors and sentence structure can be improved. E.g.

“Nevertheless, there are currently no nutritional guidelines in Latvia for women during lactation”.

Sentence 271-275

Grammatical errors and sentence structure can be improved. E.g.

“To increase awareness of women's nutritional needs during lactation period, national authorities of Latvia could adopt dietary guidelines that encourage women to consume more of milk & dairy products (preferably organically produced), to increase conjugated linoleic acid level in human milk, potentially providing further health benefits to the breastfed infant.”

Conclusion

Query a further statement on the need for further guidelines to support women in conclusion.

Sentence 277-278

Grammatical errors and sentence structure can be improved. E.g.

Add “the” before “trans fatty”

Sentence 278-179

Grammatical errors and sentence structure can be improved. E.g.

“Vaccenic and elaidic acid were the most predominant trans fatty.....”

Sentence 300

Grammatical errors.

Change declare to declares

Author Response

We would like to thank the reviewer for the detailed comments and suggestions provided for the improvement of the manuscript (nutrients-1345236). We believe that the comments have identified important areas which required improvement. After completion of the suggested edits, the revised manuscript has benefited from an improvement in the overall presentation and clarity. Below, you will find a point-by-point description of how each comment was addressed in the manuscript.  Original comments in boldface, responses in regular typeface.

Abstract

Sentence 1-3:

Good summarizing title, however, no reference that this study was conducted in Latvia. Query changing to:

“trans Fatty Acids in Human milk. Association with Dietary Habits during the Lactation Period in Latvia”

Sentence 10-11:

“mainly” does not fit the theme of this sentence. Query remove and replace with predominately.

Sentence 11-12:

This sentence structure does not make sense. Conjunctions are required to allow the sentence to flow. Query replace with:

“…. Was to determine the trans fatty acid level in human milk among lactating women from Latvia, and to evaluate how maternal dietary habits affect the trans fatty acid composition of human milk.

Sentence 16-17:

As above. Query replace with:

“Overall, the dietary intake of trans fatty acids ….”

Sentence 17-18:

As above. Query replace with:

“The Total trans fatty acid level in the human milk samples was 2.40 % ± 0.60 %.”

Sentence 18-19:

Poor sentence structure and lack of proper English. Query replace with:

“This study reports the highest vaccenic acid and lowest rumenic acid levels recorded in human milk samples when compared to previous studies (references)”

Sentence 19-20:

This sentence structure does not make sense. Query replace with:

“The composition of trans fatty acids found in human milk was associated with maternal dietary habits.”

Sentence 20-21:

Additional words used that are not required.  Remove “the”

A higher elaidic acid, vaccenic acid and total trans fatty acid level in human milk was found among the participants with a higher milk & dairy product intake.

Sentence 22-23:

Incorrect grammar. As two measures are reported level needs to be plural e.g. levels.

“Meat & meat product intake was associated with a higher vaccenic acid and total trans fatty acid levels in human milk”

Sentence 23-24:

This sentence structure does not make sense. Query replace with:

“A moderate association was also established between maternal trans fatty acid intake and the total trans fatty acid level in human milk.”

We accepted the above suggested recommendations and made necessary corrections.

Introduction

General:

Further introduction and evidence are required surrounding the relevance/benefits of trans fatty acids in breastmilk. This is not currently discussed.

We added more information regarding relevance/benefits of the trans fatty acids in human milk (Lines 54 to 65).

Sentence 29-31:

“first half of year” is overly wordy. Please rewrite e.g., “human milk is the only nutrient source during this time” – you have already referenced that you are talking about this first six months of life.

Please also correct grammar surrounding percentages e.g 15-20%

Sentence 31-32

Sentence can be refined e.g.“Nutrients for human milk are derived from maternal body stores and those absorbed directly from the maternal diet”

Sentence 32-33

This sentence structure does not make sense. Conjunctions are required to allow the sentence to flow.

Query replace with:

“Therefore, a well-balanced diet during lactation is critical both for the mother and breastfed infant”.

Additionally, the word critical suggests that serious consequences will occur if the diet is not optimal. Query replace with a word that does not have a causative relationship e.g. important.

Sentence 35-37

This sentence has a double negative “already previously”. Please select one e.g., “previously reported”

This sentences grammar can also be improved. Query replace with:

“For example, we have already previously reported that fish intake reflects on the docosahexaenoic acid level in human milk among lactating women from Latvia.”

Sentence 47

Incorrect grammar. “the’ not required before health

Sentence 47-48

Sentence flow can be improved. Query replace “The” with “This”.

Sentence 49-50

Poor word choice. “On the opposite” is an odd choice of words. Query replace with: opposingly.

Grammatical errors with poor sentence flow. Query replace with:

“On the opposite Opposingly, conjugated linoleic acid is associated with health related benefits, for example such as an antiatherogenic effect for XXX [7]”

We accepted the above suggested recommendations and made necessary corrections.

Who is an antiatherogenic affect beneficial for? Mum or baby or both? Are there any other specific benefits of trans fatty acids in breastmilk? 

We decided to remove information regarding antiatherogenic affect as we reread the information source (beneficial to humans, not specified that for mothers & infants). But we added different information sources that declare specific positive benefits for the infant of natural trans fatty acids in the human milk (Lines 59 to 65).

Sentence 47-48

Grammatical errors with poor sentence flow. “The” required to start the sentence. Query move “currently” (and change to current) before general position e.g. “The current general position currently is that human body…”

Sentence 67-68

Sentence flow can be improved. Query replace with:

“As breastmilk is the sole source of nutrition for 15-20% of infants in the first six months of life, the quality of human milk is of great importance.”

Sentence 68-71

Grammatical errors. Conjunctions are required to allow the sentence to flow. Query add “the” after affects e.g. “…..affects the trans fatty acid composition….”

We accepted the above suggested recommendations and made necessary corrections.

Materials and methods

General:

How many study appointments occurred?

Where was the research conducted? E.g., where were the research facilities or were home visits conducted?

Sentence 73-74

Did other women sign up for the study?

Did anyone drop out of the study?

We added this information in the Materials & Methods section (Lines 92 to 93, 95 to 109).

Grammatical errors. Conjunctions are required to allow the sentence to flow.

e.g. “…..during the period between January and till December year  2020.

Sentence 74

We accepted the above suggested recommendations and made necessary corrections.

Were there any exclusion criteria?

Yes, they were. Information was already provided (Lines 90 to 91).

Sentence 78

How many months (maximum) postpartum were women eligible to take part? Did postpartum age affect the results in any way?

We wanted to collect human milk samples only from women up to six months postpartum, but due to Covid-19 pandemic we faced difficulties to obtain sufficient number of samples and therefore postpartum or maternal age was not defined as a criteria. However, analyzing the data, we did not find any statistically significant impact regarding trans fatty acid level in human milk and time postpartum or maternal age (p-value >0.05).

Sentence 86-87

Poor word choices and sentence structure. Query replace with:

“During the study, participants had were asked to complete a self-administered 72-hour food diary and provide a and to obtain a 10mL pooled human breastmilk sample (hindmilk expressed from various breastfeeding sessions) of ~10 ml within next within 24 hours of their first appointment (unsure if this is correct or not??). They were also asked to complete a food frequency questionnaire (FFQ).

We accepted the above suggested recommendations and made necessary corrections.

When did each part of the study take part?

How was hind milk determined? More information required here.

We provided more detailed information in the Materials & Methods section (Lines 93 to 153).

Note food frequency questionnaire can be abbreviated to FFQ for further references. Please change in remaining text.

We accepted the above suggested recommendations and made necessary corrections.

Sentence 88

Remove sentence – add self-administered to sentence 86-87 as above.

Sentence 88-89

The same thing has been said twice. Please refine sentence e.g.

“ No dietary restrictions were applied and participants were encouraged to consume their regular diet., participants were able to consume ad libitum diet.”

Sentence 89-90

Sentence flow can be improved. Query replace with:

“Volume measures (handful, teaspoon etc.) were provided to participants to could be used to help complete the food diary.”

Sentence 90-94

Reshuffle sentence, add conjunctions, and remove the word “Unfortunately,” as it provides unnecessary judgment. Query replace with:

Unfortunately, The current Latvian food composition database lacks dietary data regarding on the fatty acid composition of different foodstuff. Therefore, the Finnish Food Composition Database Fineli [12] was used to calculate participants energy value, and nutrients intake, including trans fatty acids among the study participants.”

What nutrients were calculated?

Nutrients listed in the sentence..

We accepted the above suggested recommendations and made necessary corrections. Nutrients were listed in the sentence (Lines 149 to 150).

Sentence 94-96

Sentence can be refined to improve readability. Query replace with:

“If participants were using dietary supplements were used during the study period, nutritional information was sourced from of them was taken from the manufacturers’ website and included in nutrient calculations the calculations of total nutrient intake.”.

We accepted the above suggested recommendations and made necessary corrections. 

Sentence 97

When did they complete the FFQ?

More information is required about the timeline of your methods.

We provided more detailed information (Lines 110 to 132).

Sentence 91-99

Sentence conjunctions required. Start sentence with “The”

Sentence 99-100

Unsure of what this sentence is saying? What categories were chosen and why?

What do you mean by median intake values?

We accepted the above suggested recommendations and made necessary corrections.  We also provided more detailed information about FFQ categories to improve the clarity (Lines 114 to 132).

Sentence 101-103

How was this information collected?

When was it collected?

Query would demographics be a better word than characteristics?

We accepted the above suggested recommendations and made necessary corrections.  We also provided more detailed information to improve the clarity about determined demographic characteristics (Lines 136 to 140).

Sentence 105-106

Sentence conjunctions required. Query add “the” before “trans fatty acid …”

What solvents were used for extraction?

Query reword sentence – currently difficult to decipher if solvent or ”transesterified” occurred first.

We accepted the above suggested recommendations and made necessary corrections.  We also provided more detailed information regarding trans fatty acid composition analysis to improve the clarity (Lines 155 to 209).

Sentence 106-111

Very long sentence. Please refine.

Missing references for standards.

Sentence 111-116

Incomplete sentence. “The” required to start sentence.

Sentence 117-119

Grammatical errors.

Change “was” to “were”

How was the relative proportion of each trans fatty acid was determined?

We accepted the above suggested recommendations and made necessary corrections.  References to the standards were added (Lines 202 to 204) and more detailed information regarding trans fatty acid determination was provided (Lines 204 to 209).

Sentence 122-123

“The” not required before MS excel

MS abbreviation has not been described. Please write in full.

Sentence 124-126

Grammatical errors.

Add “the” before trans fatty acid

Results 129-131

Grammatical errors and the format between BMI groups differs.

“the” not required before body mass index.

Write BMI groups in the same format e.g.,” (n=55)” or “12 participants”

Remove word “but” unsure what you are trying to say. Places judgment on these participants.

Sentence 131-132

Query report percentage rather than number of participants.

Sentence 132-133

Almost is used to start two sentences in a row.

Query report percentage rather than number of participants.

We accepted the above suggested recommendations and made necessary corrections.  

Sentence 134-136 (table 1)

Table needs to be across one page. Table is hard to read. Query add lines between topics.

To improve the clarity of data, we divided information into two tables.

Sentence 137-139 (figure 1)

Query improve figure title. Is “among the participants” required? Who else would it be? What was the time frame of data collected from?

This figure provides various information that is important to the study, however, the figure is hard to read. Can you show this information in a different way that is easier to read? 

We improved the Figure title. We agree that information in the Figure is not easy to understand, but we did not find any better solution how to visualize the FFQ data.

If a key is being used, please refer to this as a key. 

Unfortunately, we did not understood what is meant with the key. Can the Reviewer provided more detailed explanation? Thank you in advance!

Does the table show the diet per day or across a month?

We added clarification to the title of table 3 that data are median values from 72-hour food diary (therefore median values from three days).

Sentence 138-139

What are important food products?

These have not been introduced in the introduction. If reporting in results this needs to be included in the introduction. 

We have stated what are the most important food product group of trans fatty acids in the diet (Lines 44 to 53).

Sentence 140-142

Poor sentence structure with both past and present tenses are used. I am not sure what you are trying to say ?dairy was consumed more than cereals?

Sentence can be refined to improve readability.

Is rarely the best word choice? Would it be more accurate to compare food groups according to frequencies that were questioned?

For example, XX% of women consumed starchy foods like cereals, cereal products, and potatoes daily. were consumed 140 rarely (~60 % of the participants). Opposingly XX% of women consumed milk and milk products daily. but milk & dairy products – almost every day by only one quarter of the participants.”

We accept the  the above suggested recommendations and made necessary corrections.  

Where are cereals and grains on the figure? Is this starchy foods?

Where necessary, we added the clarification that starchy foods are cereals, cereal products, potatoes.

Sentence 142-144

Grammatical errors. Sentence can be refined to improve readability.

Query replace with:

“Approximately 20% of the participants were completely avoiding avoided milk & dairy products during the lactation period. The main reason for this was due to mostly noting the reason – infant cow’s milk protein allergies for the infant.”

What percentage of avoidance was due to allergies?

We accept the  the above suggested recommendations and made necessary corrections.  We added information regarding the percentage of the participants avoiding milk & dairy products due to the infant having cow's milk protein allergy (Line 233 to 236).

Sentence 146-147

Remove “-“ to improve readability replace with “were”.

As above is rarely the best word choice?

Sentence147-149 and 149-151

Why are the reported statistics approximately? Can you record an accurate statistic?

 Sentence 149-151

You have already said this data is from the FFQ – do you need to repeat this?

Sentence 152

“the” required to start sentence.

Sentence 155-156 (table 2)

Is “among the participants” required in title?

We accept the  the above suggested recommendations and made necessary corrections. 

What guidelines were used? ?references

We used national guidelines and guidelines at the European level and they are listed in the Table 3.

The table is hard to read. Query add lines between topics to increase readability.

We added lines between the topics to increase the readability.

Can energetic value be simply written as energy?

We rewritte energetic value to energy

Why are 2-3 guidelines reported (even when the same numbers)? E.g., total fat

National (Latvian) and other guidelines at European level differ. We decided to report all nutritional guidelines.

Sentence 156 -176

Gap required between table and next paragraph

This whole paragraph needs to be reordered. It would make more sense to report results in groups e.g. not meeting energy carbohydrate and fibre requirements then higher amounts recorded for sugar and fat. Lastly meeting protein requirements.

Sentence 157-159

This seems to be two sentences in one. Please split.

Grammar and word choice can be improved, as per comment above.

Sentence 160-161

Reference required.

Sentence 161-164

Grammar, word choice, and sentence structure can be improved.

Remove “but”. Poor word choice,

% should be written directly after number e.g. 4%, 1%, and 0.5%

Sentence 164-166

Remove “among participants” – who else would it be?

How low? reference to guidelines what % was met.

Combine this sentence with below sentence - you say the same thing twice.

Sentence 167-170

Sentence conjunctions required. Add “the” before trans fatty acid intake

Sentence 170-173

Sentence structure can be improved. No need for bullet points. Query replace with:

“Maternal trans fatty acid intake was significantly higher among those with higher intakes of milk & dairy products (ρ = 0.372, p = 0.003) and sweets & bakery goods (ρ = 0.305, p = 0.017).”

Sentence 174-176

“On the opposite” is an odd phrase. Query reword to “Opposingly a significantly lower intake of ....”

Sentence 178-179

Sentence conjunctions required and grammar errors. American language used. Query replace with:

The median ± interquartile range, as well as and minimal–maximal values of trans fatty acids level in participants human milk samples among study participants are summarizsed in Table 3.

Sentence 181-182

Sentence structure can be improved. Query replace with: “Vaccenic acid was the most predominant ...”

% should be written directly after number e.g. 70%

Sentence 182-183

The not required before individual.

Is human milk required in this sentence?

Sentence 183-184

You have said level three times in this sentence

Is human milk required in this sentence?

Please reword sentence.

Sentence 185-186

How were they correlated?

Sentence 188-189

Is human milk required in this sentence?

Sentence 189-191, 189-191, and 191-193

Is human milk required three times in these sentences?

Sentence 191-193

Remove “also” – informal language.

We accept the  the above suggested recommendations and made necessary corrections. 

Discussion

Limitations and advantages of the study is not discussed. Query include in discussion? 

In the Discussion section we added limitations/advantages (like no recent similar studies provided in the Europe, therefore no data to compare, small sample size etc.) (Lines 298 to 302, 311 to 319).

Sentence 196-198

Sentence structure can be improved. Query replace with:

“This study shows a significantly lower trans fatty acid intake compared to previous research investigating lactating women”.

We accept the  the above suggested recommendations and made necessary corrections.

Sentence 199

Were all of these studies conducted with the same methodology?

Doe the low participant numbers impact these studies validity?

We clarified that no recent similar studies have been provided in the Europe, therefore no data to compare, small sample size etc.) (Lines 298 to 302, 311 to 319).

Sentence 203-205

Sentence structure can be improved. E.g. “Research from Croatia shows similar results to this study, previously identifying that ....dietary sources...”

Sentence 206-207

Sentence structure can be improved. Add “the” before total trans fatty acid ….

Sentence 207-209

Also not required to start sentence.

Query “studies” instead of researchers to increase formality of writing.

Sentence 209-211

As above query “studies” instead of researchers to increase formality of writing.

Sentence 212-213 (table 5)

“the” required before trans fatty acid

Sentence 214-216

Various grammar errors. Query replace with:

“Not all individual trans fatty acids are reported in the table, therefore the total trans fatty acid level may be higher than the total sum of individual trans fatty acids reported in the table.

 Sentence 217-218

Sentence structure can be improved e.g. “previous studies have indicated that vaccenic acid can be converted to rumenic acid during lactation”

Sentence 218-220

Sentence structure can be improved e.g.

“Nevertheless, the conversion rate is low with less than 10% of the rumenic acid in the human coming from the endogenous synthesis of vaccenic acid”

Sentence 220-221

Sentence structure can be improved e.g.

This could explain why no significant correlations between the vaccenic acid and rumenic acid level in human milk was found.

Sentence 222-225

Sentence structure can be improved e.g.

“Vaccenic acid is the predominant trans fatty acid found in animal origin fat [4]. Therefore, the strong positive association in this study between vaccenic acid levels and the intake of meat & meat products and milk & dairy products was not surprising.”

Sentence 225-227

Grammatical errors and sentence structure can be improved. E.g.

Although it should be noted that the overall, milk & dairy product and meat & meat product intake among the participants was not high with the majority of participants consuming these foods once a week or less”.

We accept the  the above suggested recommendations and made necessary corrections. 

Was the amount of food consumption recorded e.g. only consumed once per week but large amounts?

No, food amounts were not recorded in FFQ, only in 72-hour food diary.

Sentence 227-229

Grammatical errors and sentence structure can be improved. E.g.

“Approximately 20% of participants completely avoided milk & dairy products during lactation period. This was predominately due to infants having a cow’s milk protein allergy.”

Sentence 230-235

Grammatical errors and sentence structure can be improved.

e.g. “Precht & Molkentin (1999) [19] have also observed higher vaccenic acid levels in human milk among German lactating women with a higher consumption of milk and dairy products. Opposingly studies from the Netherlands suggest that a maternal diet rich in meat and dairy products (especially organically produced) is associated with a lower elaidic acid/vaccenic acid ratio in human milk (ratio of 1.27 for conventional diets versus ratio of 0.86 for diet rich in organically produced meat and dairy products). 

Sentence 236

Remove “even”. Does not add anything to sentence.  Replace with “a”

Remove “- only 0.29” with “of 0.29”

Sentence 240-242

Grammatical errors and sentence structure can be improved.  E.g.

“….. contain traces of elaidic acid, with previous studies showing the elaidic acid level of human milk can increase with heat treatments”

Sentence 243-245

Grammatical errors and sentence structure can be improved. E.g.

“Higher rumenic acid levels in human milk (0.34 %) have been reported among lactating women who are predominantly consuming meat and dairy products of organic origin compared to women on conventional diets (0.25 %) [23].”

Sentence 246-248

Grammatical errors and sentence structure can be improved. E.g.

“Data was not collected to determine if participants consumed conventional or organically produced ruminant products in this study”

“but we speculate that participants were mostly consuming conventionally produced meat & meat products, milk & dairy products”.

We accept the  the above suggested recommendations and made necessary corrections. 

Can you say this? Is there any data from similar cohorts in Latvia to suggest this?

No, unfortunately. We are the first researchers evaluation fatty acid composition in human milk among lactating women in Latvia.

Sentence 248-250 and 250-252

Grammatical errors and sentence structure can be improved. E.g.

“Within this study the FFQ results suggested that women less frequently consumed milk & dairy products as well as meat & meat products. This could potentially explain why lower rumenic acid levels are reported when compared to data from other countries”

Sentence 253-255

Grammatical errors and sentence structure can be improved. E.g.

Replace “we also report” with ‘this study also reports that the trans fatty acid level in human milk is directly linked to the maternal dietary intake of trans fatty acids.”

Sentence 255-257

Grammatical errors and sentence structure can be improved. E.g.

However, it seems that the trans fatty acid level in human milk among lactating women in Latvia is more closely related to the intake of ruminant fat intake (milk & dairy products and meat & meat products).

Sentence 257-259 and 259-262

Grammatical errors and sentence structure can be improved. E.g.

“This differs to previous studies that have found that trans fatty acid levels were related to the intake of sweets and bakery items in Polish women (reference) and sweets, bakery goods, fried food, dairy products, margarine, and sausages in Croatian women (reference).”

Sentence 262-263

Grammatical errors and sentence structure can be improved. E.g.

Query “studies” instead of researchers to increase formality of writing.

Query change “found” to “observed a”

Add “the” before “maternal intake”

Sentence 264-265

Grammatical errors and sentence structure can be improved. E.g.

Add “an” before “important”

Sentence 266-269

Grammatical errors and sentence structure can be improved.

Remove “are mostly consuming” – replace ”predominantly consume trans fatty acids from natural food sources”

“thanks to the national legislation rules in Latvia [11], women are able to choose industrially produced foodstuff without high amount of trans fatty acids who can have adverse effect both on the mother and the infant”. Was this asked in the study or an assumption?

We accept the  the above suggested recommendations and made necessary corrections. Regarding this sentence - “thanks to the national legislation rules in Latvia [11], women are able to choose industrially produced foodstuff without high amount of trans fatty acids who can have adverse effect both on the mother and the infant” - it is a statement based on mandatory rules in Latvia and in other Europen countries that declare maximum permitted amount of trans fatty acids in food.

Sentence 270-272

Grammatical errors and sentence structure can be improved. E.g.

“Nevertheless, there are currently no nutritional guidelines in Latvia for women during lactation”.

Sentence 271-275

Grammatical errors and sentence structure can be improved. E.g.

“To increase awareness of women's nutritional needs during lactation period, national authorities of Latvia could adopt dietary guidelines that encourage women to consume more of milk & dairy products (preferably organically produced), to increase conjugated linoleic acid level in human milk, potentially providing further health benefits to the breastfed infant.”

We accept the  the above suggested recommendations and made necessary corrections. 

Conclusion

Query a further statement on the need for further guidelines to support women in conclusion.

More information provided (Lines 378 to 383).

Sentence 277-278

Grammatical errors and sentence structure can be improved. E.g.

Add “the” before “trans fatty”

Sentence 278-179

Grammatical errors and sentence structure can be improved. E.g.

“Vaccenic and elaidic acid were the most predominant trans fatty.....”

Sentence 300

Grammatical errors.

Change declare to declares

We accept the  the above suggested recommendations and made necessary corrections. 

Round 2

Reviewer 3 Report

A clear difference can be seen between this edition and the original manuscript. The readability and scientific soundness has improved significantly. Further clarification of this research area and the methodology used has improved the quality of the research. To further improve the manuscript please see the below recommendations (minor).

Sentences 54-57: References are required to support these statements.  

Sentence 60-62: query moving reference to the end of this sentence

Sentence 33-34: Change “for” to “of”

 Sentence 92-93: sentence structure can be improved to increase readability. Query replace with  “In total, 70 participants completed the study. Initially, 108 participants met the study inclusion criteria, however, 38 dropped out.”

Sentence 96-98: sentence structure can be improved to increase readability. Query replace with:

If they met the study inclusion criteria, participants were invited to the research centre, where study materials and an “overview of the study was provided”.

Note: I am unsure what you mean by vague questions – are these your research questions or the participant's questions?

Sentence 103-104: Query is this sentence required. If required sentence structure can be improved to increase readability. Query replace with:

“Women were able to complete the study in a location convenient for them (for example, at home).

Sentence 104-106: Sentence structure can be improved to increase readability. Query replace with:

“Participants were asked to complete a self-administered 72-hour food diary and provide at least 10 mL of pooled human milk sample within a 24 hour period (milk expressed from the feeding breast after the end of nursing).

Sentence 114-116: “the researcher” is not required.

Sentence 135-139: Sentence structure can be improved to increase readability. Query replace with:

“Demographic characteristics such as maternal age; body mass index; parity; time postpartum; child sex, birth weight and length; and breastfeeding pattern (exclusive or partial) was collected using a self-administered questionnaire undertaken during the four day study period.”

 Sentence 141-143: Sentence structure can be improved to increase readability. Query replace with:

“A second visit was arranged after participants had complete the required tasks”

Sentence 143-146: What do you mean by study materials? Is this referring to the FFQ and food diary?

Sentence 155: sentence structure can be improved to increase readability. Query replace with “All samples were analysed within two months of arrival at the laboratory.”

Sentence 178-179: Query error in glass tube size. Did you mean 1.5mg?

Sentence 184-189: Could fractions be named “Fraction one” “fraction two” etc to improve grammar.

Sentence 196-200: Sentence structure can be improved to increase readability. Query replace with:

“The flow rate of the helium carrier gas was 1.0 mL min-1. The initial oven temperature of 100 °C was held for two minutes, then increased to 180 °C at 4 °C min-1, 210 °C at 2°C min-1, 223 °C at 0.8 °C min-1, and 240 °C at 20 °C min-1 and held for another ten minutes (total time 64.1 minutes).”

Sentence 200: Query error in time? Did you mean 61.1 minutes?

Sentence 206-208: Change “was” to “were”

Sentence 219-221: Sentence structure can be improved to increase readability. Query replace with:

“According to 219 body mass index calculations, 55 participants had a normal body weight, 12 participants were overweight, and 3 participants were obese.”

Sentence 224: Do you mean height measurements for adults? Length would be incorrect to calculate BMI in adults.

Sentence 232-233: Sentence structure can be improved to increase readability. Query replace with:

“Approximately 20% of the participants completely avoided milk & dairy products during the lactation period.”

Sentence 233-234: Sentence structure can be improved to increase readability. Query replace with:

“The main reason for this was infant cow’s milk protein allergies (10% of the participants).”

Sentence 234-235: Sentence structure can be improved to increase readability. Query replace with:

“Other reasons were vegetarian or vegan dietary patterns (4% of the participants)”

Sentence 235-236: Sentence structure can be improved to increase readability. Query replace with:

“The remaining participants (6%) did not state why milk & dairy products were avoided.”

Sentence 239-241: Sentence structure can be improved to increase readability. Query replace with:

“Half of the participants consumed sweets & baked goods seldomly and 60% consumed salty snacks & fast foods seldomly.”

Sentence 241-242: Sentence structure can be improved to increase readability. Query replace with:

Most of the participants (56%) avoided soft drinks, but caffeine-containing drinks or herbal teas were mostly consumed on daily basis (26% and 41% of the participants, respectively).

Sentence 243-244: Sentence structure can be improved to increase readability. Query replace with:

“Most of the participants (79%) reported avoidance of alcohol during the lactation period.”

Sentence 287-288: Sentence structure can be improved to increase readability. Query replace with:

“The content of trans fatty acids in natural products and therefore the intake of naturally occurring trans fatty acids is relatively constant [4,13].”

Sentence 289-291: Sentence structure can be improved to increase readability. Query replace with:

“On the other hand, the intake of industrially produced trans fatty acids within the European Union has been gradually decreasing due to the regulation rules setting the maximum permitted amount of trans fatty acids in food products [13–14,19].”

Sentence 291-293: Sentence structure can be improved to increase readability. Query replace with:

“Due to the new regulations, industrially produced food products have been reformulated and ingredients such as partially hydrogenated fats replaced, mainly with palm oil [13].”

Sentence 294: Sentence structure can be improved to increase readability. Query replace with:

“Also, this study shows a low trans fatty acid intake among lactating women (Table 5).”

Sentence 298-301: Sentence structure can be improved to increase readability. Query replace with:

Overall, the trans fatty acid intake among lactating women in Europe has not been comprehensively analysed in the past decade. Few studies, that have been conducted, differ in methodology and have low participant numbers (<100 participants) (Table 5).

Sentence 301-302: Impossible is a strong word. Query reword.

Sentence 307-310: Sentence structure can be improved to increase readability. Query replace with:

Researchers from Croatia [21] found results, identifying that dominant dietary sources of trans fatty acids among lactating women were sweets & bakery goods and milk & dairy products.”

Sentence 315-319: Sentence structure can be improved to increase readability. Query replace with:

“Referring to the results from this study and taking into account recent legislation changes in the European Union, it can be speculated that the trans fatty acid intake, as well as trans fatty acid levels in human milk among lactating women in Europe, has likely decreased in the past few years.”

341-344: Sentence structure can be improved to increase readability. Query replace with:

“ A Study from the Netherlands [22] suggest that a maternal diet rich in meat and dairy products (especially organically produced) is associated with a lower elaidic acid/vaccenic acid ratio in human milk (ratio of 1.27 for conventional diets versus the ratio of 0.86 for a diet rich in organically produced meat and dairy products).”

Sentence 339-345:
How do these studies compare? A further critical review of this paragraph with reference to this study's results is required. 

Author Response

We would like to thank the reviewer for the second review provided for the last improvements of the manuscript (nutrients-1345236). Below, you will find a point-by-point description of how each comment was addressed in the manuscript.  Original comments in boldface, responses in regular typeface.

Sentences 54-57: References are required to support these statements.

We added reference in the Line 56.

 Sentence 60-62: query moving reference to the end of this sentence

 We moved the reference to the end of the sentence (Line 62-63).

 Sentence 33-34: Change “for” to “of”

 “For” changed to “of” in Line 33.

 Sentence 92-93: sentence structure can be improved to increase readability. Query replace with  “In total, 70 participants completed the study. Initially 108 participants met the study inclusion criteria, however, 38 dropped out.”

 We changed the sentence to improve readability.

 Sentence 96-98: sentence structure can be improved to increase readability. Query replace with:

If they met the study inclusion criteria, participants were invited to the research centre, where study materials and an “overview of the study was provided”.

Note: I am unsure what you mean by vague questions – are these your research questions or the participants questions?

 We corrected the sentence. However, participants were not invited to the research centre, but met with the researcher at any convenient place for the participant (for example, at participant’s home).

We corrected sentence – “If a woman met the study inclusion criteria, she met with the researcher (in a location convenient for the participant, for example, at home). During the meeting, study materials and a sample container for human milk collection were provided. If necessary, the researcher answered the participant's questions.”.

 Sentence 103-104: Query is this sentence required. If required sentence structure can be improved to increase readability. Query replace with:

“Women were able to complete the study in a location convenient for them (for example, at home).

 We replaced the sentence to improve the readability.

 Sentence 104-106: Sentence structure can be improved to increase readability. Query replace with:

Participants were asked to complete a self-administered 72-hour food diary and provide at least 10 mL of pooled human milk sample within a 24 hour period (milk expressed from the feeding breast after the end of nursing).

 We replaced the sentence to improve the readability.

 Sentence 114-116: “the researcher” is not required.

 In Line 114-116 there are no words “the researcher”. Maybe you mean Line 118? We removed these words from the Line 118.

 Sentence 135-139: Sentence structure can be improved to increase readability. Query replace with:

“Demographic characteristics such as maternal age; body mass index; parity; time postpartum; child sex, birth weight and length; and breastfeeding pattern (exclusive or partial) was collected using a self-administered questionnaire undertaken during the four day study period.”

 We replaced the sentence to improve the readability.

 Sentence 141-143: Sentence structure can be improved to increase readability. Query replace with:

“A second visit was arranged after participants had complete the required tasks”

 We replaced the sentence to improve the readability.

 Sentence 143-146: What do you mean by study materials? Is this referring to the FFQ and food diary?

 Yes, we added explanation to the Lines 143 to 146 – “A second visit was arranged after participants had complete the required tasks. During a second meeting (on-site), the researcher collected all the study materials (the completed 72-hour diary, FFQ etc.) and a human milk sample from each participant.

 Sentence 155: sentence structure can be improved to increase readability. Query replace with “All samples were analysed within two months of arrival at the laboratory.

 We replaced the sentence to improve the readability.

 Sentence 178-179: Query error in glass tube size. Did you mean 1.5mg?

 We corrected word “tube” to “vial”. Correct volume is 1,5 mL.

 Sentence 184-189: Could fractions be named “Fraction one” “fraction two” etc to improve grammar.

 Yes, of course. We renamed them to fraction one, two etc.

 Sentence 196-200: Sentence structure can be improved to increase readability. Query replace with:

“The flow rate of the helium carrier gas was 1.0 mL min-1. The initial oven temperature of 100 °C was held for two minutes, then increased to 180 °C at 4 °C min-1, 210 °C at 2°C min-1, 223 °C at 0.8 °C min-1, and 240 °C at 20 °C min-1 and held for another ten minutes (total time 64.1 minutes).”

 We replaced the sentence to improve the readability.

 Sentence 200: Query error in time? Did you mean 61.1 minutes?

 We rechecked the in-house method for this analysis BIOR-T-012-131-2011 and recalculated the total time by ourselves. It is 64.1 minutes.

 Sentence 206-208: Change “was” to “were”

 “Was” was replaced to “were” in Line 205.

 Sentence 219-221: Sentence structure can be improved to increase readability. Query replace with:

“According to 219 body mass index calculations, 55 participants had a normal body weight, 12 participants were overweight, and 3 participants were obese.”

 We replaced the sentence to improve the readability.

 Sentence 224: Do you mean height measurements for adults? Length would be incorrect to calculate BMI in adults.

 Yes, in the Line 223 we corrected “length” to “height”.

 Sentence 232-233: Sentence structure can be improved to increase readability. Query replace with:

“Approximately 20% of the participants completely avoided milk & dairy products during the lactation period.”

 We replaced the sentence to improve the readability.

 Sentence 233-234: Sentence structure can be improved to increase readability. Query replace with:

“The main reason for this was infant cow’s milk protein allergies (10% of the participants).”

 We replaced the sentence to improve the readability.

 Sentence 234-235: Sentence structure can be improved to increase readability. Query replace with:

“Other reasons were vegetarian or vegan dietary patterns (4% of the participants)”

 We replaced the sentence to improve the readability.

 Sentence 235-236: Sentence structure can be improved to increase readability. Query replace with:

“The remaining participants (6%) did not state why milk & dairy products were avoided.”

 We replaced the sentence to improve the readability.

 Sentence 239-241: Sentence structure can be improved to increase readability. Query replace with:

“Half of the participants consumed sweets & baked goods seldomly and 60% consumed salty snacks & fast foods seldomly.”

 We replaced the sentence to improve the readability.

 Sentence 241-242: Sentence structure can be improved to increase readability. Query replace with:

Most of the participants (56%) avoided soft drinks, but caffeine containing drinks or herbal teas were mostly consumed on daily basis (26% and 41% of the participants, respectively).

 We replaced the sentence to improve the readability.

 Sentence 243-244: Sentence structure can be improved to increase readability. Query replace with:

“Most of the participants (79%) reported avoidance of alcohol during the lactation period.”

 We replaced the sentence to improve the readability.

 Sentence 287-288: Sentence structure can be improved to increase readability. Query replace with:

“The content of trans fatty acids in natural products and therefore the intake of naturally occurring trans fatty acids is relatively constant [4,13].”

 We replaced the sentence to improve the readability.

 Sentence 289-291: Sentence structure can be improved to increase readability. Query replace with:

“On the other hand, the intake of industrially produced trans fatty acids within the European Union has been gradually decreasing due to the regulation rules setting the maximum permitted amount of trans fatty acids in food products [13–14,19].”

 We replaced the sentence to improve the readability.

 Sentence 291-293: Sentence structure can be improved to increase readability. Query replace with:

“Due to the new regulations, industrially produced food products have been reformulated and ingredients such as partially hydrogenated fats replaced, mainly with palm oil [13].”

 We replaced the sentence to improve the readability.

 Sentence 294: Sentence structure can be improved to increase readability. Query replace with:

“Also, this study shows a low trans fatty acid intake among lactating women (Table 5).”

 We replaced the sentence to improve the readability.

 Sentence 298-301: Sentence structure can be improved to increase readability. Query replace with:

“Overall, the trans fatty acid intake among lactating women in Europe has not been comprehensively analysed in the past decade. Few studies, that have been conducted, differ in methodology and have low participant numbers (<100 participants) (Table 5).

 We replaced the sentence to improve the readability.

 Sentence 301-302: Impossible is a strong word. Query reword.

 We rephrased the sentence – “Therefore, data can only be compared in the context of the significant reduction in the trans fatty acid content of the food due to legislation changes and therefore resulting in lower consumption of trans fatty acids among lactating women in Europe.” (Lines 301 to 305).

 Sentence 307-310: Sentence structure can be improved to increase readability. Query replace with:

Researchers from Croatia [21] found results, identifying that dominant dietary sources of trans fatty acids among lactating women were sweets & bakery goods and milk & dairy products.

 We replaced the sentence to improve the readability.

 Sentence 315-319: Sentence structure can be improved to increase readability. Query replace with:

“Referring to the results from this study and taking into account recent legislation changes in the European Union, it can be speculated that the trans fatty acid intake as well as trans fatty acid levels in human milk among lactating women in Europe has likely decreased in the past few years.”

We replaced the sentence to improve the readability.

 341-344: Sentence structure can be improved to increase readability. Query replace with:

“A Study from the Netherlands [22] suggest that a maternal diet rich in meat and dairy products (especially organically produced) is associated with a lower elaidic acid/vaccenic acid ratio in human milk (ratio of 1.27 for conventional diets versus the ratio of 0.86 for a diet rich in organically produced meat and dairy products).

 We replaced the sentence to improve the readability.

Sentence 339-345: How do these studies compare?

 We supplement the sentence “Although milk & dairy products, as well as meat & meat products intake was not high, this study reports a lower elaidic/vaccenic acid ratio in human milk of 0.29 compared to the study from the Netherlands [22].” (Lines 344 to 347).